



# Neural networks for data assimilation of surface and upper-air data in Rio de Janeiro

Vinícius Albuquerque de Almeida[1], Haroldo Fraga de Campos Velho[2], Gutemberg Borges França[1], and Nelson Francisco Favilla Ebecken[3]

[1]Laboratory for Applied Meteorology - Federal University of Rio de Janeiro
[2]National Institute for Space Research
[3]Civil Engineering/COPPE - Federal University of Rio de Janeiro

**Correspondence:** Vinícius Albuquerque de Almeida (vinicius@lma.ufrj.br)

**Abstract.** The practical feasibility of neural networks models for data assimilation using local observations data in the WRF model for the Rio de Janeiro metropolitan region in Brazil is evaluated. Surface and multi-level variables retrieved from airport meteorological stations are used: air temperature, relative humidity, and wind (speed and direction). Also, 6-hour forecast from WRF high-resolution simulations are used – domain centered in the Rio de Janeiro city with nested grids of 8 and 2.6 km. Peri-

ods of 168h from 2015-2019 are used with 6h and 12h assimilation cycles for surface and upper-air data, respectively, applied to 6-hour forecast fields. The observed data (interpolated to grid points close to airport locations and influence computed in its surroundings) and short-range forecasts are used as input for training model and the 3D-Var analysis on 6-hour forecast fields for each grid point is used as target variable. The neural network models are built using two different approaches: WEKA multilayer perceptron model and TensorFlow's deep learning implementation. The year of 2019 is used as an independent dataset

for forecast validation from the trained models. Results employing 6-hour forecast fields with neural network models are able to emulate the 3D-Var results for surface and multi-level variables, with better results for the NN-TensoFlow implementation. The main result refers to CPU time reduction enabled by the neural networks models, reducing the data assimilation CPU-time by 121 times and 25 times for NN-TensorFlow and NN-WEKA, respectively, in comparison to the 3D-Var method under the same hardware configurations.

# 1   Introduction

Numerical weather prediction is an initial-value problem as stated in the beginnings of the twentieth century by Bjerknes (1904). Therefore, given an estimate of the present state of the atmosphere (initial conditions), and appropriate surface and lateral boundary conditions, the model simulates (forecasts) the atmospheric evolution (Kalnay, 2002). Currently, data assimilation methods are used by operational centers to determine as accurately as possible the state of the atmospheric flow – also

for hydrology, ocean circulation, atmospheric pollutant predictions – (Talagrand, 1997). The determination of such state is essential, sophisticated, and demands high computational effort (Daley, 1991). Several methods have been developed since the 1950s to tackle this problem. See Daley (1991), Talagrand (1997), Zupanski and Kalnay (1999), and Kalnay (2002) for reviews on the subject.



The Data Assimilation Research Testbed (DART) group (Anderson et al., 2009) provides a wide range of applications for
data assimilation, such as: atmospheric reanalyses (Kistler et al., 2001; Uppala et al., 2005; Compo et al., 2006); estimation
of the value of existing or hypothetical observations (Khare and Anderson, 2006; Zhang et al., 2004); prediction of efficient
flight paths for planes that release dropsondes (Bishop et al., 2001) and assessing the potential impact of a new satellite
instrument before it is built or launched (Mourre et al., 2006); guide model development by estimating model parameter values
more consistent with observations (Houtekamer and Mitchell, 2001; Aksoy et al., 2006); ocean applications (Keppenne and
Rienecker, 2002; Zhang et al., 2005), land surface (Reichle et al., 2002), cryosphere (Stark et al., 2008), biosphere (Williams
et al., 2005), and chemical constituents (Constantinescu et al., 2007); among other applications of different areas of engineering,
geophysics, space weather, oil extraction, economy, medicine, and social sciences.

The traditional approaches for data assimilation applied to numerical weather prediction usually are both computer and
time-consuming requiring super computers for daily use in operational centers. Therefore, in the last two decades efforts have
been made in order to overcome the resource restrictions of data assimilation speeding-up the process without loss of quality.
In this context, neural networks have been applied to emulate several data assimilation methods, such as: Kalman filter (Härter
and Campos Velho, 2008; Härter and de Campos Velho, 2010), ensemble Kalman filter (Cintra and Campos Velho, 2018),
particle Filter (Furtado et al., 2008), and variational methods (Furtado et al., 2011; Wu et al., 2021). Data assimilation by
neural networks has been applied to space weather (Härter et al., 2008), 2D shallow water for ocean circulation (Sambatti
et al., 2018), urban air pollution (Casas et al., 2020), hydrology (Boucher et al., 2020), medicine (Arcucci et al., 2020).

Our research is to employ neural network to emulate a variational scheme applied to data assimilation for a regional at-
mospheric model. The Weather Research and Forecasting model (WRF) is a community atmospheric modeling system, and
its development and capabilities are the result of the contributions of a host of individuals and institutions over the years
(Skamarock et al., 2019a). This model contains a limited-area three-dimensional variational data assimilation (3D-Var) system
applicable to both synoptic and mesoscale numerical weather prediction (Barker et al., 2004).

The aim of this analysis is to assess an artificial neural network's capacity to emulate the 3D-VAR approach for surface
and upper-air data in the Rio de Janeiro metropolitan area, Brazil. Similar to Cintra and Campos Velho (2018) and Zhu et al.
(2019), neural network is trained to emulate a more computational intensive data assimilation method. However, for higher
dimension dynamical system, it is necessary to consider a strategy for computing the observation influence. Different from
Cintra and Campos Velho (2018), the Cresmann's interpolation (Cressman, 1959) is applied in this paper as the observation
influence operator.

The present article is part of a sequence of studies related to nowcasting under development and implementation by the
Laboratory for Applied Meteorological at the Federal University of Rio de Janeiro, following Almeida (2009), Silva et al.
(2016), França et al. (2016), França et al. (2018), and Almeida et al. (2020). All these studies encompass researches based on
artificial intelligence methods for weather forecasts, mainly for high-impacting phenomena for aviation. The development is
carried out with the cooperation of the Department of Aerial Space Control (DECEA: *Departamento do Controle de Espaco
Aéreo*, a division of the Brazilian Air Force.





## 2 WRF: Limited area meteorological model

The WRF model is a next-generation mesoscale numerical weather prediction system designed for both atmospheric research
and operational forecasting applications. It features two dynamical cores, a data assimilation system, and a software archi-
tecture supporting parallel computation and system extensibility. The effort to develop WRF began in the latter 1990s and
was a collaborative partnership of the National Center for Atmospheric Research (NCAR), the National Oceanic and At-
mospheric Administration (represented by the National Centers for Environmental Prediction (NCEP) and the Earth Sys-
tem Research Laboratory), the U.S. Air Force, the Naval Research Laboratory, the University of Oklahoma, and the Fed-
eral Aviation Administration (FAA). Please refer to the WRF Users Guide and the Technical Note document available at
http://www2.mmm.ucar.edu/wrf/users/ for completeness of the 3D-Var implementation present at WRF (Ska-
marock et al., 2019b).

The WRF model solves a set of equations modeling the state and evolution of the atmosphere, including: (i) conservation of
momentum; (ii) thermodynamic energy conservation; (iii) mass conservation; (iv) geopotential relation; and (v) the equation
of state. Also, several physical processes are parameterized (e.g. short and longwave radiation transfer, surface modeling,
turbulence, cumulus convection, cloud microphysics, and precipitation). These ones are too small, too brief, too complex, too
poorly understood, or too computationally costly to be explicitly represented.

## 3 Data Assimilation Methods

Data assimilation is a special type of inverse problem (Nakamura and Potthast, 2015) by periodically determining initial con-
dition combining observations with previous estimation/forecasting. The inverse solution can be calculated applying different
techniques based on regularization theory (Tikhonov and Arsenin, 1977; Campos Velho et al., 2006), Kalman filter (Schillings
and Stuart, 2017; Murakami and Hasegawa, 1993), and artificial neural network (Krejsa et al., 1999). Another approach is the
nudging or Newtonian relaxation, adding a forcing term given by the difference between the computed field and observations
multiply by a nudging parameter (Hoke1 and Anthes, 1976) – for modern version of this approach see Auroux and Blum
(2008). However, the most of operational centers for weather prediction has adopted variants of variational methods or Kalman
filter.

Here, data assimilation techniques are applied to a 3D limited area meteorological model: 3D-Var method, and artificial
neural network. The later approach is trained to emulate the 3D-Var by using two frameworks of neural networks. Brief
descriptions of the employed data assimilation methods are given in this Section.

### 3.1 3D-Var approach

Among various data assimilation methods, variational approaches have been widely used in meteorology, specifically the
method 3D-Var. In the 3D-Var approach, a cost function (1) is proportional to square differences between the analysis ($x^a_{n+1}$)
and both the background ($x^b_{n+1}$) added to the difference with observations ($y^o_{n+1}$) (Kalnay, 2003). The analysis field is com-





puted by the direct minimization of such function. Important to notice that the co-variance matrices for both the background

($B$) and observation error ($R$) are considered in the minimization process. An operator $H$ maps the gridded analysis to the observation space for comparison against the observation vector $y^o$. The analysis $x^a$ is computed by minimizing the cost function ($J$) expressed as:

$$J(x) = \frac{1}{2} \left\{ [y^o - H(x)]^{\mathrm{T}} R^{-1} [y^o - H(x)] + [x - x^b]^{\mathrm{T}} B^{-1} [x - x^b] \right\} \tag{1}$$

where $R$ is the covariance matrix of the sensor errors, and $B$ is the background covariance matrix.

Here the CV3 option of WRFDA is used in order to represent $B$, which is the NCEP background error covariance. It is estimated in grid space by what has become known as the NMC method (Parrish and Derber, 1992). The statistics are estimated with the differences of 24 and 48-hour GFS forecasts with T170 resolution, valid at the same time for 357 cases, distributed over a period of one year. Both the amplitudes and the scales of the background error have to be tuned to represent the forecast error in the estimated fields. The statistics that project multivariate relations among variables are also derived from

the NMC method. More information is available at the WRF User's Page at https://www2.mmm.ucar.edu/wrf/users/docs/user_guide_v4/v4.4/users_guide_chap6.html#_Generic_BE_option:.

The 3D-Var approach consists in processing observed information in a temporal window (typically from 1 h before the analysis time to 1 h after) over a spatial domain. After this process a subset of the observed data is retrieved that will be assimilated in a previous forecast grid by the minimization of a cost function. The 3D-Var approach is used as implemented in

the Data Assimilation component of the WRF framework. The basic ideas of variational data assimilation and specifically the WRF Data Assimilation (WRFDA) system is discussed in-depth in Barker et al. (2012).

### 3.2 Neural network technique

Artificial neural networks is a branch of artificial intelligence belonging to the class of *machine learning* (ML) algorithms – see Rosenblatt (1958), Hopfield (1982), Rumelhart et al. (1986) and Haykin (1999). An ANN is an arrangement of several

connected processing units. These units are called *neurons*, where the weighted inputs can or not be combined with a bias to feed a nonlinear *activation function*: $\varphi(z)$ – such as ReLU function (Rectified Linear Unit), tanh function, or Gaussian one. ANN can be roughly classified into two groups: supervised and unsupervised neural networks. For the first one, there is a reference data set to be used to identify the connection weights ($w_{ij}^k$: value for connecting the $i$-th input with $j$-th neuron from $L$-th layer). A very employed supervised ANN is the *multi-layer perceptron* (MLP).

The MLP-NN is a supervised network, and it typically consists of a set of layers: the input layer (one or more inputs), one or more hidden layers, and the output layer (one or more outputs). The well known back-propagation algorithm is a standard procedure to determine the connection weights – the process is named as *the training or learning* phase (Haykin, 1999) (Section 4.3). Figure 1 shows a representation of fully connected neural network with two hidden layers – the number of hidden layers denotes how *deep* is the neural network. The back-propagation algorithm is a gradient method, and it can

also be applied to convex functions, as the ReLU function, under the concept of subdifferential (Rockafellar, 1970). Therefore,





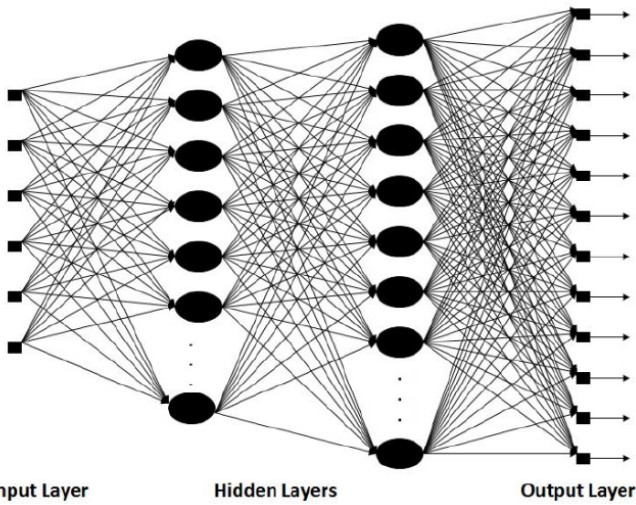

**Figure 1.** Fully connected artificial neural network with two hidden layers.

subdiff$\{$ReLU$\}(x)_{x<0} = 0$, subdiff$\{$ReLU$\}(x)_{x>0} = 1$, and subdiff$\{$ReLU$\}(x)_{x=0} = \zeta$, where $\zeta \in [0,1]$, but in general the value is chosen as $\zeta = 0$ to have a more sparse matrix.

As mentioned, a standard back-propagation algorithm is a gradient driven algorithm, where the inverse of Hessian matrix is approximated by $\eta I$, being $\eta$ the learning ratio and $I$ the identity matrix. Several strategies can be employed to improve the back-propagation. One scheme is by adding a *momentum* term:

$$
\begin{aligned}
\Delta w_{ij}^k(n+1) &= \eta \, \delta(n) \, y_j^k(n) + \alpha \, \Delta w_{ij}^k(n) &\qquad(2)\\
\Delta w_{ij}^k(n) &= w_{ij}^k(n) - w_{ij}^k(n-1) &\qquad(3)
\end{aligned}
$$

where $n$ is the iteration or epochs, $\delta(n)$ is the gradient of the loss function: square difference between the reference values (training set) and the ANN outputs, $y_j^k(n)$ is the output of the $j$-th neuron in $k$-th hidden layer, and $\alpha$ is a momentum constant. The momentum factor shows the effect of past weight changes on the up date in in weight space. This term allows to deal with a larger learning ratio without oscillations (Haykin (1999)). Depending on the value of learning ratio, the matrix $(\eta I)$ can be considered as an *approximate inverse* for the Hessian matrix (Flavell (1977)). Here, the Hessian is defined as $H(n) \equiv \nabla \delta(n)$, being $\delta(n)$ the gradient in Eq. 2. The approximate inverse is a matrix following the property:

$$
\|I - (\eta I) H(n)\| < 1 \,, \qquad(4)
$$

for some matrix norm. Assuming that the learning ratio $\eta$ is in agreement with condition 4 above, and if the iteration procedure 2 follows the Lipschitz condition: $\|f(x) - f(y) < c\|x - y\|$, for $c < 1$, the process converges to the fixed-point (Conte and De Boor, 1980) — see Section 5.3 in the book.

Two frameworks for designing neural networks for data assimilation are employed here by WEKA and TensorFlow software packages. The WEKA package embraces several ML algorithms. The TensorFlow software with focus on deep learning



neural networks. Arcucci and co-authors has applied deep learning strategy for data assimilation (Arcucci et al., 2021) for low dimensional order dynamical system. For the present application, a dynamical system with much higher dimension order – a 3D limited atmospheric model – is the goal of our application.

### 3.2.1 WEKA

WEKA stands for Waikato Environment for Knowledge Analysis, and it is a comprehensive software resource created to illustrate the ideas of the book *Data Mining: Practical Machine Learning Tools and Techniques* (Witten et al., 2017). The software package is available as Java source code – see the internet link: http://www.cs.waikato.ac.nz/ml/weka. The software includes full, working, state-of-the-art implementations of many popular ML schemes, to be used for practical data mining or for research. Finally, it contains a framework, in the form of a Java class library, for supporting applications employing embedded machine learning and even the implementation of recent learning schemes.

WEKA provides implementations of learning algorithms, easily applicable for a user dataset. It also includes a variety of tools for transforming datasets, such as the algorithms for discretization and sampling. A data analyst can preprocess a dataset, feed it into a learning scheme, and analyze the resulting machine learning model, evaluating its performance—all without writing any computer code at all. Also, the software includes methods for the main data mining problems: regression, classification, clustering, association rule mining, and attribute selection. Getting to know the data is an integral part of the work, with many data visualization facilities, and data preprocessing tools are provided. All algorithms take their inputs in the form of a single relational table, these data can be read from a file or generated by a database query (Witten et al., 2017). Here, the multi-layer perceptron neural network is applied to emulate the 3D-Var scheme for data assimilation.

### 3.2.2 TensorFlow

TensorFlow is an interface for expressing machine learning algorithms, with an implementation for executing such algorithms. Its origins refer to the Google Brain project started in 2011 to explore the use of very-large-scale deep neural networks, both for research and for use in Google's products. Based on the experience of other projects within the group (e.g. DistBelief), including a more complete understanding of the desirable system properties, and requirements for training and using neural networks, TensorFlow was built. A second-generation system for the implementation and deployment of large-scale machine learning models were also developed. This is an open-source software library for machine learning applications codified by Python, C++, CUDA programming languages. Several tasks can be carried out, but it has been mainly applied for training and inference of deep neural networks.

TensorFlow takes computations described using a dataflow-like model and maps them onto a wide variety of different hardware platforms, ranging from running inference on mobile device platforms, such as Android and iOS, to modest-sized training and inference systems – using single machines containing one or many GPU cards – to large-scale training systems running on hundreds of specialized machines with thousands of GPUs. The system is flexible and it can be used to express a wide variety of algorithms, including training and inference algorithms for deep neural network models. The TensorFlow has been used for conducting research and for deploying machine learning systems into production across more than a dozen





areas of computer science and other fields, including speech recognition, computer vision, robotics, information retrieval, natural language processing, geographic information extraction, and computational drug discovery. The TensorFlow API and a reference implementation were released as an open-source package under the Apache 2.0 license in November, 2015, and are available at http://www.tensorflow.org (Abadi et al., 2015).

## 4 Experiments and data

### 4.1 Study area

The study area is the metropolitan area of Rio de Janeiro city and its surroundings (Fig. 2), which is roughly centered at latitude 22°55'44.3"S and longitude 43°24'21.1"W. The most import airports in the region are located in Figure 2 identified by their International Civil Aviation Organization (ICAO) codes: Santos Dumont Airport (SBRJ), Galeão International Airport (SBGL), Santa Cruz Air Force Base (SBSC), Jacarepaguá Airport, and Afonsos Air Force Base (SBAF).

### 4.2 Observational data

Each airport is responsible for local hourly routine and special reports surface observations of several meteorological parameters as surface wind (direction and speed), visibility, significant weather, cloud cover, air and dewpoint temperature, and station pressure. Besides, the SBGL airport has an upper-air (or sounding) station for producing regularly atmospheric soundings twice a day, the atmospheric profile of pressure, air and dewpoint temperature, relative humidity, and wind (direction and speed), from the surface up to more than 25 km.

### 4.3 Experiments

Experiments with 1-week data assimilation are performed using the WRF model from 2015 and 2019, starting at February 1st with 168h for time-integration (seven days). The data assimilation is performed in 6-hour forecast fields. This method is carried out every 6 hours for surface variables (air temperature, relative humidity, and wind direction and speed) at all airport locations and surroundings under its influence, and every 12 hours for upper-air variables (air temperature, relative humidity, and wind direction and speed) at SBGL location for all vertical grid points of the model.

In our numerical experiments, the WRF model is integrated into a two-nested grids (8 and 2.6 km) with 45 levels in vertical direction, generating hourly outputs from the surface and pressure-level variables. Regarding the parameterizations, the following options were chosen: Microphysics – WRF Single–moment 3 (Hong et al., 2004), Cumulus – Grell–Freitas Ensemble Scheme (Grell and Freitas, 2014), Radiation – Dudhia Shortwave Scheme (Dudhia, 1989)/ RRTM Longwave Scheme (Mlawer et al., 1997), Planetary Boundary Layer – Yonsei University Scheme (YSU) (Hong et al., 2006), and Land-Surface model – Unified Noah Land Surface Model (Tewari et al., 2016).

The experiment steps consisted of

**Figure 2.** Domain and computational grids. The labels SBSC, SBAF, SBJR, SBRJ, and SBGL are located at the airports in the metropolitan area of Rio de Janeiro.




(i) generation of analysis field from observations and 6-hour forecast (background) field using the 3D-Var data assimilation technique in WRF Data assimilation (WRFDA) module technique from 01/02 to 08/02 00Z with surface data assimilation every 6 hs and upper-air data assimilation every 12hs, which was repeated for the same period of 168hs for years from 2015 to 2019;

(ii) generation of pseudo-observations based on 5-grid points centered in the airport stations evaluating the impact of these observations on the surroundings using a modified version of the method presented by Kalnay (2003) in Chapter 5, section 5.2.1, defined as:

$$f_i = \frac{\sum_{k=1}^{K} q_{ik}(f_k^O - f_i)}{\sum_{k=1}^{K_i} q_{ik}} \tag{5}$$

where $f_i$ is the value of the field estimate at grid point-$i$, $K$ is the number of the available observations, $f_k^O$ is the $k$-th observation surrounding the grid point $i$. The weights $q_{ik}$ represent the magnitude of influence of the $k$-th observation surrounding the grid point-$i$ to the final estimate at grid point-$i$. Here the definition by Cressman (1959) is applied as follows:

$$q_{ik} = \frac{R_n^2 - r_{ik}^2}{R_n^2 + r_{ik}^2} \quad for \quad r_{ik}^2 \le R_n^2 \tag{6}$$

$$q_{ik} = 0 \quad for \quad r_{ik}^2 > R_n^2 \tag{7}$$

The code to execute the pseudo-observation calculation is available at Almeida et al. (2022) in a file called *functions.py* under the *util* directory.

(iii) observations, 6-hour forecast (background) field and analysis computed for each grid point (for observation and pseudo-observations) are merged in order to obtain a single dataset for each analysis time;

(iv) a pre-processing is executed for data cleansing and normalization;

(v) the period from 2015 to 2018 is used for training, year 2019 is used for model test and evaluation;

(vi) training is performed using MLP models built by Weka and Tensorflow frameworks with observations and background of each grid point being used as neural networks inputs and 3d-var analysis as neural networks target as supervisor of nn output;

(vii) an evaluation is performed comparing the results for the data assimilation process by the 3D-Var data and neural networks.

The trained models are available at Almeida et al. (2022) under the *models* directory with the names *nn-tf.h5* and *nn-weka.model* for the tensorflow model and the weka model, respectively. These models are intended to replace the WRFDA component on the WRF Modeling System flow chart (see Chapter 1 of the WRF User's Guide available at https://www2.mmm.ucar.edu/wrf/users/docs/user_guide_v4/v4.4/users_guide_chap1.html#WRF_Modeling_System). That is, each grid point on the background field generated by the WRF *Real* component is updated directly by the neural networks outputs.





## 5   Results

This section presents the results of the trained neural network models applied to the test dataset during the period February 1st-8th, 2019, our independent dataset. In addition, a GitHub repository including all of the relevant files and information on the procedure utilized in our study is provided for repeatability and applicability in other experiments.

The training process of neural networks is an optimization problem. However, the configuration of neural network architecture depends also on to determine multiple parameters – hyper-parameters – in order to find appropriate topologies for the studied problem. Table 1 presents the final topology and other pertinent information regarding the neural networks trained in TensorFlow and Weka. Many topologies with different configurations were tested and explored, but for the sake of simplicity only the configurations that produced the best results for the current application are presented. Also, the training process is

a time-consuming task and Table 2 presents the consumed CPU time for neural networks training – both in TensorFlow and Weka – using the dataset explored in this work.

    Figure 3 presents the 2-m air temperature map on Feb 1st, 2019 12 UTC for the control field, 3D-Var and by networks trained in the TensorFlow (NN-TensorFlow) and Weka (NN-Weka). This date was randomly chosen from the test dataset to be used here to illustrate the performance of data assimilation, performing the analysis by NN in comparison to 6-hour forecast from a

field without data assimilation (control) and 6-hour forecast field from an analysis with 3D-Var data assimilation (target).

    The control map (Fig. 3a) presents the smoothest field, with highest temperatures at the metropolitan area. The 3D-Var method (Fig. 3b) shows a intensification of higher temperatures at the metropolitan area, in accordance to the surface observed data (not presented, but available at: https://www.redemet.aer.mil.br) which showed 2-m air temperature higher than control field at station locations. The field shows the expected behavior for data assimilation outputs, where an analysis

field is generated by the optimum adjust between the background field and the available observations. Figures 3c-d present the results from the NN-TensorFlow and NN-Weka, respectively. Interesting to note that both neural network methods are able to identify higher temperatures compared to the control field, showing sensibility to the observation data, this means, learning the process of adjustment between the background and the observed fields. Compared to the 3D-Var, the target variables in the training process, both NN methods show higher values indicating a larger perturbation in the surroundings of the station

locations. As explained in section 2.1, the 3D-Var method computes an innovation matrix in the observation grid and then interpolates data back to the model grid using the $H$ observation operator. In the NN training, the process is different, the influence of observations is computed in the model grid for training and no interpolation is necessary to generate the analysis field. Therefore, the behavior of the 3D-Var field for 2-m air temperature map (3b) could be related to an interpolation effect.

    In Figure 3, a subjective analysis was displayed between the neural networks results and 3D-Var applied to a 6-hour forecast

field, but an objective quantification of the differences between the 6-hour forecast field with data assimilation and the observation field is necessary to evaluate whether the differences between neural networks and 3D-Var indicates a decrease of quality in the data assimilation process or not, or even an improvement.

    Figure 4 presents the temperature error map on Feb 1st, 2019 12 UTC (same day analyzed in Figure 3 and randomly chosen from the test dataset for illustration of the present discussion) for 3D-Var, NN-TensorFlow and NN-Weka in relation





to the observed map. The errors show that the neural networks are able to reduce the overall error in the study area (in the surroundings of the station data). The reason for such improvement is not clear at a first glance, since the 3D-Var is used as a target variable and, thus, the neural networks are expected to be limited in its quality by the quality in the output reference (to what it is compared in the cost function evaluation).

As explained in the analysis of Figure 3, one of the possible explanations for this improvements is in the computation of
the impact of the station data on the surroundings. The 3D-Var performs an interpolation of the grid data to the observation data for innovation matrix computation, and then interpolates once more to the original grid domain for the final field. The dataset for neural networks training was created in an opposite direction (see Experiments description in Section 4). Pseudo-observations were created in a region of 5-grid points radius centered in each station location computing the influence of the station data already in the grid domain. This way the neural network was already trained in the final grid and the possible errors
for interpolation and smoothing existent in 3D-Var were not inserted in the trained neural networks. Important to observe that differently from 3D-Var (Fig. 4a) and NN-TensorFlow (Fig. 4b), showing an overall underestimation of the 2-m air temperature field, the NN-Weka (Fig. 4c) results show regions of overestimation close to the Southeastern part of the metropolitan area of Rio de Janeiro, where there is a higher density of stations. As a final remark in the analysis of Figure 4, the error map of the NN models resembles the pattern in 3D-Var – showing the capacity of the learning algorithms – and can even improve the
performance of the 3D-Var data assimilation approach, largely used in many operational centers.

Besides the surface data, profiles of air temperature, humidity, and wind (speed and direction) were also assimilated in this research, retrieved from the SBGL upper-air station. Figure 5 presents the average air temperature error profile at SBGL from Feb 1st, 2019 up to Feb 8th, 2019 for 6-hour forecast from a field without data assimilation (control), 6-hour forecast field with 3D-Var data assimilation (target), 6-hour forecast field with NN-TensorFlow data assimilation, and 6-hour forecast field
with NN-Weka data assimilation, in relation to the observed profile from surface up to 20 km, in the stratosphere. Important to remind that, differently from the surface data, profile data at SBGL is available only at two-times a day, at 00 and 12 UTC, and thus, Figure 5 presents an average of 14 profiles (2 soundings × 7 days).

From Figure 5, it is possible to identify two different regions with larger errors: at surface up to 7.5 km (surface up to middle troposphere) and in the stratosphere (17.5 to 20 km). The overall effect of the data assimilation process is to reduce the mean
profile error, mainly on the above mentioned regions, where the control output presents larger errors (e.g. under 2.5 km, the average error in control output reaches a maximum of 3 K, whereas the data assimilation outputs does not exceed 2 K). The analysis of the NN model results show a close accordance to the 3D-Var curve, with a mean error reduction (for large part of the profile, with some exceptions close to the tropopause), and slightly better results for NN-TensorFlow (NN-TF) in comparison to NN-Weka. Important to note that the region of the upper troposphere and lower stratosphere is characterized by control
files with better results than the assimilation curves, but the errors are all under 1 K, what might be neglected in a confidence interval.

Figure 5 allowed an overall (subjective) analysis of the average error profile, but an objective statistical analysis of the profile from data assimilation in the test dataset might be helpful to evaluate a certain degree of confidence of the data assimilation process and the impact of replacement of 3D-var method for NN approaches. Table 3 presents a summary of the statistics of





the assimilation methods applied to the test dataset (from Feb 1st to Feb 8th, 2019). Analyzing the mean error, there is a 22% (from 0.61 to 0.48) mean error reduction from the 3D-Var to NN-Tensorflow, and an 29% increase (0.61 to 0.79) to NN-Weka. The root mean square error (RMSE) is important to be analyzed, because it gives an extra information in comparison to the mean error, since it avoids the positive and negative values cancellation. The results resembles what the behavior of the mean error, with NN-TensorFlow having the lowest RMSE values among the three methods. Important to note that the maximum mean error and RMSE do not exceed 0.79 and 1.12, respectively, what is relatively low, considering the uncertainty from the instrument. The standard deviation (SD) provides an information regarding the level of uncertainty of the data assimilation process. The maximum SD is observed for NN-TensorFlow with 0.59 K, and the minimum for NN-Weka with 0.44 K. In general, the analysis of the statistics presented in Table 3 show an overall accordance of the NN methods to reproduce the performance of the 3D-Var.

The discussions presented up to this point of the paper showed a good accordance of the NN models for data assimilation process in comparison to the reference method (3D-Var). Therefore, an important question to analyze is whether if there is plausible justification for the replacement of a traditional and validated method as 3D-Var for a new one based on neural networks. Although a slightly better performance was observed for the NN models, these do not seem statistically significant to answer such question. In order to address an answer for such question, lets to looking at the CPU time consumption for the assimilation methods in a single assimilation cycle and for the test period – see Table 4. The 3D-Var method in our experiments took 8 seconds for a single data assimilation cycle, and it had overall consumption of 2 minutes and 27 seconds. The NN methods had a single cycle (total) CPU time of 0.04 seconds / 1.21 seconds, and 0.21 seconds / 5.88 seconds for NN-TensorFlow and NN-Weka, respectively. Proportionally, the overall CPU time reduction was around 121 times and 25 times for NN-TensorFlow and NN-Weka, respectively. Another advantage of the NN methods is not necessary a powerfull hardware computer resources as the 3D-Var, as implemented in WRF framework.

The code repository is publicly available at Zenodo (Almeida et al., 2022). Both Weka and Tensorflow trained models and the algorithm for pseudo-observation creation are available for reproducibility. Here the v4.2 of WPS, WRF and WRF-DA were used. The source code for WRF (including the DA component) and WPS are available at https://github.com/wrf-model.

## 6 Conclusions and Final Remarks

This paper studies the ability of neural networks to emulate the 3D-Var method implemented in the WRF data assimilation framework, for surface and vertical profile data assimilation applied to 6-hour forecast fields. Experiments with 168 hs for time integration of the WRF model were carried out on the terminal area of Rio de Janeiro between 2015 and 2019, with observations obtained at five different airport locations.

Our results can be summarized as follows:

1. As expected, the data assimilation process applied to 6-hour forecast fields had an overall effect of reduction of the errors in relation to observed values, through the innovation vector added to the background field;







**Figure 3.** Temperature map of (a) 6-hour forecast field (without data assimilation), (b) 6-hour forecast field with 3D-Var and 6-hour forecast field with neural networks - trained in (c) TensorFlow and (d) Weka - on Feb 1st, 2019 12 UTC.



**Figure 4.** Temperature error map of (a) 3D-Var and neural networks - trained in (b) TensorFlow and (c) Weka - applied to 6-hour forecast fields in relation to the observed map on Feb 1st, 2019 12 UTC.



**Figure 5.** Mean temperature profile difference of 6-hour forecast (from a field without data assimilation), 6-hour forecast with 3D-Var and neural networks - trained in TensorFlow and Weka - in relation to the observed profiles between Feb 1st and 8th, 2019 at SBGL.



**Table 1.** Final topology and other characteristics of neural networks in TensorFlow and Weka experiments.

| Parameter | NN-TensorFlow | NN-Weka |
|---|---|---|
| Version | 2.0.0 | 3.9.3 |
| Number of Layers | 3 | 2 |
| Number of hidden units (each layer) | 30 | 25 |
| Activation function (hidden layers) | ReLU | Sigmoid |
| Activation function (output) | Unthresholded linear | Unthresholded linear |
| Optimizer | Adam[1] | Backpropagation[2] |
| Learning rate | 0.001 (default) | 0.3 (default) |
| Momentum | 0.9 (default) | 0.2 (default) |
| Epochs | 1000 | 500 |

[1]https://keras.io/api/optimizers/adam/

[2]https://weka.sourceforge.io/doc.dev/weka/classifiers/functions/MultilayerPerceptron.html

**Table 2.** Training time for neural networks. The time is formatted as hh:mm:ss where hh represents hours, mm represents minutes and ss represents seconds or fraction.

| Library | Time |
|---|---|
| TensorFlow | 08:57:38 |
| Weka | 09:10:05 |

2. The surface fields showed accordance between the 3D-Var method and NN on 6-hour forecast fields with overall reduction of the assimilation errors.

3. The profile data assimilation on 6-hour forecast fields also showed good accordance between the 3D-Var method and the NN models, with superior performance for the NN-TensorFlow;

4. The neural data assimilation method was 121 times and 25 times faster for NN-TensorFlow and NN-Weka, respectively, than 3D-Var technique;

5. Our results show a huge reduction of the CPU-time for the assimilation cycle, as shown in previous results (Cintra and Campos Velho, 2012; Härter and Campos Velho, 2012; Casas et al., 2020).

Important to highlight that more computational effort is needed to the 4D-Var than 3D-Var method. Using the test case for the native variational schemes in a quad-core computer, the 4D-Var method is 210 times slower than 3D-Var. Future works will investigate the performance of neural networks face on 4D-Var data assimilation, including hybrid techniques (Wang et al., 2008a, b), with an evolving background error matrix. As a final note, considering the operational centers with a relative



**Table 3.** Statistics of 6-hour forecast field with 3D-Var and neural networks data assimilation of temperature profiles between Feb 1st and 8th, 2019 at SBGL.

| Method | Mean Error (K) | RMSE (K) | SD (K) |
|---|---|---|---|
| 3D-Var | 0.61 | 1.06 | 0.50 |
| NN-TensorFlow | 0.48 | 0.99 | 0.59 |
| NN-Weka | 0.79 | 1.12 | 0.44 |

**Table 4.** CPU time for temperature assimilation on 6-hour forecast fields using 3D-Var and neural networks - trained in TensorFlow and Weka - for a single assimilation cycle and the total time elapsed performing data assimilation in the test dataset (Feb 1st to 8th, 2019). The time is formatted as hh:mm:ss where hh represents hours, mm represents minutes and ss represents seconds or fraction.

| Library | Single cycle | Total |
|---|---|---|
| 3D-Var | 00:00:08.00 | 00:02:27.00 |
| TensorFlow | 00:00:00.04 | 00:00:01.21 |
| Weka | 00:00:00.21 | 00:00:05.88 |

short time window to elaborate forecast bulletins, the reduction of CPU-time to the assimilation cycle is important for several
aspects: possibility of assimilation of a greater amount of data and/or the use of finer model computer grid resolution.

*Author contributions.* VA and HFCV developed the theoretical framework. VA developed the software package, ran and analyzed the example simulation, and prepared the paper. HFCV, GBF and NFFE supervised the work and reviewed and edited the paper extensively.

*Data availability.* Initial and boundary conditions used for WRF simulations are openly available at https://rda.ucar.edu/datasets/ds083.2/.
Surface and upper-data used in the assimilation process are openly available at https://www.redemet.aer.mil.br and also in the model reposi-
tory at https://github.com/aa-vinicius/data-assimilation-nn. Background and analysis files are available at https://github.com/aa-vinicius/data-assimilation-nn.

*Competing interests.* The authors declare that no competing interests exist.



*Acknowledgements.* This study is funded by the Department of Airspace Control (DECEA) via the Brazilian Organization for Scientific and Technological Development of Airspace Control (CTCEA) (GRANT: 002-2018/COPPETEC_CTCEA). Authors would also like to thank the

National Council for Scientific and Technological Development (CNPq, Brazil) for the research grants: HFCV (CNPq: 312924/2017-8), and GBF (CNPq: 304441/2018-0).



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
