# Peer review of "Neural networks for data assimilation of surface and upper-air data in Rio de Janeiro"

_Geoscientific Model Development, 2022_

## Referee Comment (RC2)

Title: Neural networks for data assimilation of surface and upper-air data in Rio de Janeiro
Author(s): Vinícius Albuquerque de Almeida et al.
MS No.: gmd-2022-50
MS type: Development and technical paper

Minor Reviews:

Line 40: should be "and medicine".
Line 54: "encompass researches" should be "encompasses research"
Line 61: "in the latter 1990s" sounds a bit weird
Line 80: "the most of operational centers for weather prediction has" -> "most operational centers for weather prediction have"
Line 85-95: My issue with this is equation 1 doesn't match up with the notation you are describing. You describe $x_{N+1}^a, x_{N+1}^b, and\ y_{N+1}^o$ but these are not shown in equation 1. You should describe where n and a appear if you use them, or don't mention them if you don't use them.
Line 89 and 94: co-variance vs covariance.
Line 108: "is" should be "are"
Line 110: "can or not be" sounds awkward
Line 111: "or Gaussian one." The word "one" seems weird here.
Line 112: You say there are two types (supervised and unsupervised) but you only describe one (the supervised). It would be good to add what the unsupervised method is.
Line 124: "Therefore…. Sparse matrix" needs to be reworked. Since you cite it in the line before, you could even just remove this line. If you want to keep it in, it needs a little fixing.
Line 184: This sentence needs to be improved.
Line 186: "regularly" to "regular" OR just remove "regularly" since it's not needed.
Line 190: "1-week data assimilation" what does this mean? This paragraph reads like you are doing updates every 6 (or 12) hours, with forecasts of 6 hours, and this is done over many years, but I am confused what is happening for 1 week?
Line 236: "also on to determine" I am not sure what you mean here.
Line 314: "lets to looking" should probably be "let's look"

Major Reviews:

1. Why train using the 3D-Var solution instead of real observations as the NN target variable? Was 3D-Var solution the only target variable? How does using the 3D-Var solution as the target variable compare to using the real observations? By using the 3D-Var solution as the target variable for the Neural Networks, you are estimating the 3D-Var solution and not the observations. I am a little confused at why you would do that and if it's beneficial.

2. Only one date (1 day or 1 week) as results, what if those dates are good but the rest are worse? This needs some sort of overall result from all dates. You have the whole of 2019 for testing data, does this change if you have different months? Maybe this method worked well for February 1st but not so well on other dates. I like the figures for examples that you have, they

really do show what happens at one time, but I think an overall 2019 average for each method would be nice. Maybe a nice year long line graph showing table 3 results for each day would be nice. That would really show that the results are not cherry picked (I am not saying they are, but people could come to that conclusion here without an overall year plot).

3.  This one is related to number 1. If you use the 3D-Var as a target, then your neural networks are not estimating the system, they are estimating the 3D-Var solution. In figure 4, your temperature error maps. 3D-Var will be estimating the system, so your error would be 3D-Var solution minus the real observation. Are you NNs set out like:
    1)  NN estimating 3D-Var minus real observations, or
    2)  NN estimating 3D-Var minus 3D-Var solution.

---

## Author Comment (AC2)

Dear Sir/Madam,

The authors would like to thank the editors for comments/suggestions/corrections, helping to improve the present version of the paper. We have carefully revised the manuscript. Parts of the text were rewritted and reorganized. Here, we present a brief context, following point-by-point answers for all questions.

**REFEREE-1 COMMENTS:**

**\*\* Major ones \*\***

**RC-1:** "Assimilation like 4D-Var or EnKF did requires huge computation efforts. However, the 3D-Var calculation complexity is proportion to the size of model or observations, it is usually trivial as illustrated in Table 4 (several seconds). Even handling models with larger size or with super data like remote sensing obsers, the issue could be solved through regional analysis easily. The choice of 3D-Var is faint to support the motivation."

**ANSWER-1:** Authors fully agree that 4D-Var and EnKF have a higher computational effort than 3D-Var. With 5 observations, the 3D-Var spends 8 secs. However, for assimilating an image 1024 x 1024 pixels, and supposing a linear CPU time (in fact, it isn't true for the 3D-Var, 4D-Var, and EnKF) the assimilation of such image, using the same software/hardware, this takes  - $(8/5) \times 1024 \times 1024 \sim 466$ hours ($\sim 19$ days). However, same assimilation using NN is just the numbers of number of the pseudo-observation points – in other words, the number of model grid points covering the image. For the NN, the CPU-time is exactly linear with the number of grid points. Therefore, by using TensorFlow, this takes $\sim (0.04/5) \times 1024 \times 1024 \sim 2.5$ hours.

**RC-2:** "In Figure 3 and 4: The author provides very limited samples or snapshots of analysis for testing their trained NN model, without stating the overall performance in the whole testing dataset."

**ANSWER-2:** Figures 3 and 4 are not intended to represent exhaustive testing. The figures are illustrative showing the analysis results produced with 3D-Var and 2 neural network methods emulating (both of them emulating 3D-Var scheme). Much more examples were executed with synthetic observations from our previous papers – reader can access the papers Cintra and Campos Velho (2012), and Campos Velho et al. (2022) – in the last cited paper, there is a complexity analysis showing a smalller computational complexity by using neural network.

**RC-3a:** "Page 9, line 206: only 5 airport measurements are assimilated for analysis. Meanwhile, these same data are used for generation of pseudo-observation for validating the analysis?"

**ANSWER-3a:** Yes. Five airport measurements are used to compute pseudo-observations.

**RC-3b:** "That is not the corrected way to using the measurements. Crossing validation is required. Please Check Ref: Peter Rayner. Data assimilation using an ensemble of models: a hierarchical approach., 2020, ACP."

**ANSWER-3b:** Cross-validation is a strategy used during the learning phase for both neural networks (WEKA and TensorFlow). We include a note on cross-validation strategy in the new paper version (Section 3.2). Thank you for your comment.

**RC-4:** "In Table 3, NN-TensorFlow outperforms the 3D-Var? It is not solid, afterall, 3D-Var analysis is the learning object of NN? Performance should be examined in-depth."

**ANSWER-4:** Results in Table 3 show better "Mean Error" and "RMSE" for TensorFlow than 3D-Var. However, it is not possible to have a final conclusion from the worked example. Statistics with much more examples are in our list of tasks to be carried out in future work.

**\*\* Minor \*\***

**RC-5:** "As long as they described the CPU time for assimilation in 3D-Var, NN-TF, NN-Weka in Table 4. It is essential to illustrate the size of the problem, vec x and y in Eq(1), and the solver/environment for 3D-Var and NN. Otherwise, the comparison is unfair".

**ANSWER-5:** In the new paper version, we added the number of grid points on the directions "x" and "y" (Section 4.3).

**RC-6:** "How to train the NN is unclear, what is the output actually? the analysis over the whole model domain? Or is it trained grid by grid? How many samples in their 4-year dataset?"

**ANSWER-6:** The scheme for data assimilation using NN is described in the Section 4.3 - paragraph initiated by "The experiment steps consisted of", in step-(iii): 'observations, 6-hour forecast (background) field and analysis computed for each grid point (for observation and pseudo-observations) are merged in order to obtain a single dataset for each analysis time; ...'.

We agree to write a clearer text to separate the training phase for the NN for the execution phase by applying NN:

- NN Training phase: observations and 6-hour forecast (background) field are used as inputs for the neural networks, with 3D-Var analysis used as a reference, that means, the neural network output must be the analysis for each grid point with observation or pseudo-observation.

- NN Execution phase: inputs are observation (or pseudo-observation) and background, and the output is the analysis.

The analysis is produced for each grid point with observation or pseudo-observation to be assimilated – we introduced a new text in the manuscript (Section 3.2). We use 4 cycles of data assimilation per day for 365 days, performing 1460 samples for dataset of training.

---

## Author Comment (AC3)

Dear Sir/Madam,

The authors would like to thank the editors for comments/suggestions/corrections, helping to improve the present version of the paper. We have carefully revised the manuscript. Parts of the text were rewritted and reorganized. Here, we present a brief context, following point-by-point answers for all questions.

**REFEREE-2 COMMENTS:**

**\*\* Minor Reviews \*\***

**RC-1:** Line 40: should be "and medicine".

**ANSWER-1:** The correction was implemented.

**RC-2:** Line 54: "encompass researches" should be "encompasses research"

**ANSWER-2:** The correction was implemented.

**RC-3:** Line 61: "in the latter 1990s" sounds a bit weird

**ANSWER-3:** We rewrite the sentence.

**RC-4:** Line 80: "the most of operational centers for weather prediction has" → "most operational centers for weather prediction have"

**ANSWER-4:** The correction was implemented.

**RC-5:** Line 85-95: My issue with this is equation 1 doesn't match up with the notation you are describing. You describe $x^a_{N+1}$, $x^b_{N+1}$, and $y^o_{N+1}$ but these are not shown in equation 1. You should describe where $n$ and $a$ appear if you use them, or don't mention them if you don't use them.

**ANSWER-5:** We clarified the notation in equation-1. In Section 3.2, we changed the notation. Equation 1: the state vector $x$ represent a vector where the entries are the values on all grid points. The assimilation by NN is performed over each grid point, where the inputs are the background $x^b_{ij}$ and observation (or pseudo-observation) $y^o_{ij}$, with the output being the value $x^a_{ij}$.

**RC-6:** Line 89 and 94: co-variance vs covariance.

**ANSWER-5:** We did the correction ("covariance") – Line 89.

**RC-7:** Line 108: "is" should be "are"

**ANSWER-5:** We did the correction.

**RC-8:** Line 110: "can or not be" sounds awkward

**ANSWER-8:** We rewrite the sentence.

**RC-9:** Line 111: "or Gaussian one." The word "one" seems weird here.

**ANSWER-9:** We changed for "Gaussian function".

**RC-10:** Line 112: You say there are two types (supervised and unsupervised) but you only describe one (the supervised). It would be good to add what the unsupervised method is.

**ANSWER-10:** We included some comments on unsupervised strategy for training, addressing literature to help a reader.

**RC-11:** Line 124: "Therefore.... Sparse matrix" needs to be reworked. Since you cite it in the line before, you could even just remove this line. If you want to keep it in, it needs a little fixing.

**ANSWER-11:** We follow the suggestion, and the "sparse matrix" expression was removed.

**RC-12:** Line 184: This sentence needs to be improved.

**ANSWER-12:**

**RC-13:** Line 186: "regularly" to "regular" OR just remove "regularly" since it's not needed.

**ANSWER-13:** The word "regularly" was removed.

**RC-14:** Line 190: "1-week data assimilation" what does this mean? This paragraph reads like you are doing updates every 6 (or 12) hours, with forecasts of 6 hours, and this is done over many years, but I am confused what is happening for 1 week?

**ANSWER-14:** Thank you for your commnent. The text was rewritten.

**RC-15:** Line 236: "also on to determine" I am not sure what you mean here.

**ANSWER-15:** Thank you for your commnent. The text was rewritten.

**RC-16:** Line 314: "lets to looking" should probably be "let's look"

**ANSWER-16:** We followed the suggestion.

**\*\* Major Reviews \*\***

**RC-M1:** Why train using the 3D-Var solution instead of real observations as the NN target variable? Was 3D-Var solution the only target variable? How does using the 3D-Var solution as the target variable compare to using the real observations? By using the 3D-Var solution as the target variable for the Neural Networks, you are estimating the 3D-Var solution and not the observations. I am a little confused at why you would do that and if it's beneficial.

**ANSWER-M1:** Good question. Data assimilation is a process to combine previous forecasting (*background*) with observation. Why don't we substitute predicted grid point values by observations? Because this implies a *data shock* in the simulation: There is a balance among the simulated variables, and such balance is not the same among variables into real world. Therefore, it is necessary to take into account the equilibrium among computer variables, but approximating the simulation values with the real dynamical system state. If our data assimilation (DA) method works well, but it requires a high

computational effort, and this can be a restriction that makes the method unfeasible in some applications. The latter feature is a motivation to search for an algorithm with similar numerical efficiency to the applied method but reducing the CPU time.

Therefore, the goal here is to reduce the computational effort, but with similar performance of the variational method for data assimilation process.

**RC-M2:** Only one date (1 day or 1 week) as results, what if those dates are good but the rest are worse?

**ANSWER-M2a:** We are using several years as training set, and the learning process was carried out by using cross-validation. The performance for ANNs is very good for all years (not shown). But, recovering the results from the training set could not be good enough to verify if the ANN is effective for data out of the training set. Therefore, we did 28 cycles of data assimilation with data out of the training set,

**RC-M2:** ... This needs some sort of overall result from all dates. You have the whole of 2019 for testing data, does this change if you have different months? Maybe this method worked well for February 1st but not so well on other dates. I like the figures for examples that you have, they really do show what happens at one time, but I think an overall 2019 average for each method would be nice. Maybe a nice year long line graph showing table 3 results for each day would be nice.

**ANSWER-M2b:** Yes, of course, our results do not represent exhaustive testing. Indeed, statistics with much more examples are in our list of tasks to be carried out in future work. The idea is to perform the same numerical experiments for all months in the year 2019.

**RC-M2:** ... That would really show that the results are not cherry picked (I am not saying they are, but people could come to that conclusion here without an overall year plot).

**ANSWER-M2c:** Thank you for the comment. February was the selected month because it is the harder month to do a prediction. February is the month with the strongest thunderstorms in the Rio de Janeiro city, among all months in the year — we added a comment on that in the manuscript text from climatological data (we cited the reference). With intense precipitation, February becomes the most difficult month to produce good forecasting. We include an explanation in the text justifying our choice for February month: first paragraph in Section 4.3.

**RC-M3:** This one is related to number 1. If you use the 3D-Var as a target, then your neural networks are not estimating the system, they are estimating the 3D-Var solution. In figure 4, your temperature error maps. 3D-Var will be estimating the system, so your error would be 3D-Var solution minus the real observation. Are you NNs set out like: 1) NN estimating 3D-Var minus real observations, or 2) NN estimating 3D-Var minus 3D-Var solution.

**ANSWER-M3:** As mentioned answering question 1 (M1), it is necessary to follow the dynamics in the real world, but taken into account the intrinsic balance from the computer model. Sophisticated methods for data assimilation were developed for that, but some of them are not applied yet in operational centers due to the high computational effort (for example, particle and flow filters). We cited in the introduction that our research

belongs to a set of studies with focus on nowcasting for aviation meteorology. The WRF is the model to be used for forecasting in this context, and the 3D-Var as an option for data assimilation. The computer resource for executing WRF is not a supercomputer, motivating us to investigate the application of the neural network to emulate 3D-Var.

The performance of all methods for data assimilation is computed considering the analysis (produced by 3D-Var and/or ANNs) minus observation (Figures 4 and 5).